# Compressed Video Contrastive Learning

**Yuqi Huo**[2,3,•] **Mingyu Ding**[4,•] **Haoyu Lu**[1,2] **Nanyi Fei**[2,3]
**Zhiwu Lu**[1,2,*] **Ji-Rong Wen**[1,2] **Ping Luo**[4]
[1]Gaoling School of Artificial Intelligence, Renmin University of China, Beijing, China
[2]Beijing Key Laboratory of Big Data Management and Analysis Methods
[3]School of Information, Renmin University of China, Beijing, China
[4]The University of Hong Kong, Pokfulam, Hong Kong, China
{bnhony, luzhiwu}@ruc.edu.cn
[•] Equal contribution     [*] Corresponding author

## Abstract

This work concerns self-supervised video representation learning (SSVRL), one topic that has received much attention recently. Since videos are storage-intensive and contain a rich source of visual content, models designed for SSVRL are expected to be storage- and computation-efficient, as well as effective. However, most existing methods only focus on one of the two objectives, failing to consider both at the same time. In this work, for the first time, the seemingly contradictory goals are simultaneously achieved by exploiting compressed videos and capturing mutual information between two input streams. Specifically, a novel Motion Vector based Cross Guidance Contrastive learning approach (MVCGC) is proposed. For storage and computation efficiency, we choose to directly decode RGB frames and motion vectors (that resemble low-resolution optical flows) from compressed videos on-the-fly. To enhance the representation ability of the motion vectors, hence the effectiveness of our method, we design a cross guidance contrastive learning algorithm based on multi-instance InfoNCE loss, where motion vectors can take supervision signals from RGB frames and vice versa. Comprehensive experiments on two downstream tasks show that our MVCGC yields new state-of-the-art while being significantly more efficient than its competitors.

## 1   Introduction

Recent self-supervised image representation learning approaches [He *et al.*, 2020; Chen *et al.*, 2020] have been reported to outperform supervised ones on a wide range of downstream tasks by (1) leveraging a large amount of unlabeled data available online for pre-training, and (2) designing a powerful algorithm to discriminate unlabeled samples with different semantic meanings. However, the situation for self-supervised video representation learning (SSVRL) is somewhat different: self-supervised methods still perform worse than supervised ones, since videos are extremely storage-intensive (scalability thus becomes a serious issue) and have a richer source of visual content (SSVRL thus becomes very difficult). Therefore, designing storage-and-computation-efficient as well as effective models for SSVRL remains a challenging problem that has not been well studied.

Existing state-of-the-art methods [Han *et al.*, 2020b; Tao *et al.*, 2020; Huo *et al.*, 2021] mainly focus on designing effective algorithms without considering the storage and computation costs during large-scale video self-supervised training. Particularly, in order to leverage both appearance and temporal information in the video, they exploit the optical flow as an extra view to complement the RGB stream. However, this procedure is both storage- and computation-intensive because of storing the decoded frames and computing optical flows, respectively (*e.g.*, it costs more than 100GB for

35th Conference on Neural Information Processing Systems (NeurIPS 2021).

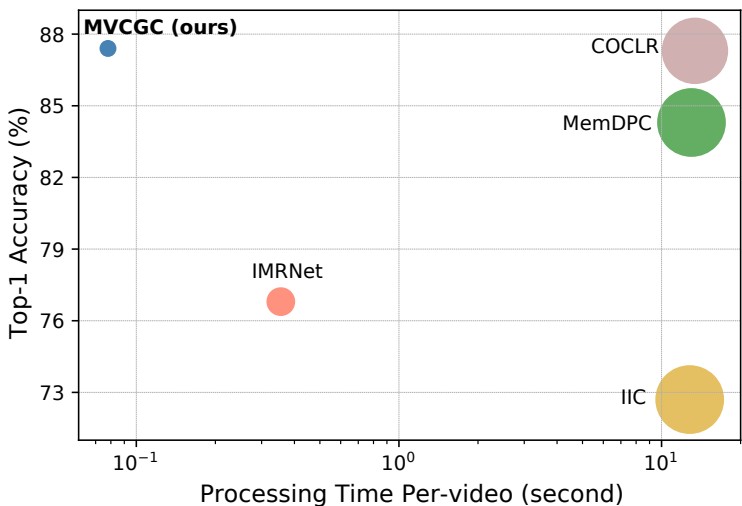

Figure 1: Processing time per-video (*i.e.*, data pre-processing and inference) and the accuracies for different methods on UCF101. Each method is represented by a circle, whose size represents the storage size occupied by the test data. MVCGC achieves the highest accuracy under a significantly reduced storage budget compared to its counterparts. More details can be found in Table 1.

frame storage and a number of days for computing optical flows [Han *et al.*, 2020b] on the UCF101 dataset [Soomro *et al.*, 2012]). This clearly hinders large-scale video self-supervised training.

Considering that videos are already stored in compressed formats to reduce storage requirements, a natural choice is to resort to on-the-fly decoding, *i.e.*, decoding frames from compressed videos during training/inference without introducing any extra storage. Moreover, since motion vectors encoded in compressed videos resemble low-resolution optical flows in describing local motions, they can be used to avoid the costly optical flow computation. Consequently, compressed videos inherently contain both static and motion information that is suitable for video representation learning. Recently, a pretext task based method called IMRNet [Yu *et al.*, 2021] leverages the standard process in CoViAR [Wu *et al.*, 2018] to decode compressed videos, but inevitably encounters the inefficiency and ineffectiveness problems: (1) The outdated CoViAR is not truly on-the-fly, *i.e.*, it re-encodes videos and stores them before decoding, which is storage and computation inefficient. (2) Motion vectors are less discriminative and thus weaker than optical flows, resulting in sub-optimal performance compared with state-of-the-art methods using optical flows, as shown in Figure 1. As a result, exploiting the compressed videos efficiently, followed by learning discriminative representations, is still challenging.

Motivated by the above observations, we propose a Motion Vector based Cross Guidance Contrastive learning approach (MVCGC), which has several appealing benefits: (1) It is able to decode RGB frames and motion vectors from compressed videos with various codecs. The storage and computation budgets are alleviated since videos can be decoded without re-encoding. (2) Our MVCGC can learn discriminative features from both RGB and motion vector streams. This is achieved by designing a cross guidance contrastive learning algorithm based on multi-instance InfoNCE loss: each sample pair is constructed by one RGB clip and one motion vector clip, and multiple positive samples can be mined by calculating the similarity in two views. Therefore, the two views can take supervision signals from each other to improve the representation quality of both streams, especially for the motion vector stream (whose learned representations are even comparable to optical flow features).

Our main contributions are three-fold: (1) We propose an efficient and effective framework called MVCGC that can learn representations from compressed videos directly. To the best of our knowledge, we are the first to exploit contrastive loss in compressed video self-supervised learning. (2) We design a novel contrastive learning algorithm to capture mutual information between RGB frames and motion vectors from compressed videos. Importantly, we even make the learned features of motion vectors as representative as those of optical flows. (3) Extensive experiments by applying the learned video representations on two downstream tasks (*i.e.*, action recognition and action retrieval) across different benchmarks demonstrate that our MVCGC achieves state-of-the-art performance while being more efficient than its counterparts (see Figure 1).

## 2 Related Work

**Self-Supervised Video Representation Learning.**   Existing self-supervised methods for video representation learning can be divided into two categories: (1) One common way to learn good representations is to define and solve pretext tasks, such as ordering frames [Misra *et al.*, 2016; Lee *et al.*, 2017; Xu *et al.*, 2019; Luo *et al.*, 2020], predicting the speed of videos [Yao *et al.*, 2020; Benaim *et al.*, 2020; Wang *et al.*, 2020], and solving space-time cubic puzzles [Kim *et al.*, 2019]. (2) Another more recent way focuses on instance discrimination, which leverages distance-based contrastive loss and distinguishes the positive samples from a group of negative ones [Oord *et al.*, 2018; He *et al.*, 2020; Chen *et al.*, 2020]. Recently, many efforts based on contrastive learning have demonstrated promising results. They either focus only on RGB frames [Han *et al.*, 2019; Zhuang *et al.*, 2020; Wang *et al.*, 2020; Kong *et al.*, 2020; Wang *et al.*, 2021] or introduce an additional optical flow view in order to achieve state-of-the-art accuracy [Han *et al.*, 2020a; Tian *et al.*, 2020; Tao *et al.*, 2020; Han *et al.*, 2020b]. Considering the heavy computational and data-storage burdens for computing the optical flow, our proposed MVCGC replaces it by introducing the motion vector from the compressed video as a new view, which significantly cuts down the computational and storage costs. Moreover, MVCGC is different from the previous approaches [Han *et al.*, 2020a; Tian *et al.*, 2020; Tao *et al.*, 2020; Han *et al.*, 2020b] in how contrastive loss and sample pairs are defined. CMC [Tian *et al.*, 2020] and IIC [Tao *et al.*, 2020] use the single-instance InfoNCE loss, resulting in neglect of hard positives. MemDPC [Han *et al.*, 2020a] and CoCLR [Han *et al.*, 2020b] construct sample pairs of contrastive loss in RGB frames and optical flows separately, ignoring the correspondence between different views. In contrast, our MVCGC constructs sample pairs between different views of different clips (*i.e.*, RGB frames of one clip and motion vectors of another clip from the same video) and applies the multi-instance InfoNCE loss. We use two views as cross guidance to train the representations of two streams simultaneously and learn mutual information between RGB frames and motion vectors.

**Compressed Video Representation Learning.**   Earlier approaches have explored the compressed video for supervised recognition [Zhang *et al.*, 2018; Wu *et al.*, 2018; Shou *et al.*, 2019], in which large-scale supervised datasets (such as ImageNet [Russakovsky *et al.*, 2015]) are used for pre-training. In contrast, we focus on self-supervised learning, *i.e.*, only unlabeled data is used in the pre-training stage. After the first exploration of using compressed video to learn "the arrow of time"[Wei *et al.*, 2018], a related work, IMRNet [Yu *et al.*, 2021], also explores compressed video self-supervised learning besides supervised learning. Our proposed MVCGC is fundamentally different from IMRNet in three key aspects: (1) Instead of focusing on designing pretext tasks as in IMRNet, our MVCGC leverages contrastive loss. (2) With the outdated MPEG-4 Part 2 codec [Huo *et al.*, 2020], IMRNet has to re-encode and store videos for data loading, whereas our MVCGC can decode compressed videos on-the-fly and save storage spaces. (3) Besides the decoded RGB and motion vector streams, IMRNet leverages an additional residual stream computed from the two streams, which has already been proved to only supply a marginal performance gain. In comparison, our MVCGC utilizes the first two streams and thus alleviates computation burdens.

## 3 Methodology

This work aims to learn robust video representations efficiently from compressed videos by designing a contrastive learning algorithm. In this section, we first review the basics of compressed video and discuss the advantages of our decoding process compared to existing counterparts. Afterward, we detail our MVCGC on how it learns effective feature representations and captures mutual information between RGB frames and motion vectors from compressed videos.

### 3.1 Basics of Compressed Video

Considering the redundancy in consecutive frames, codecs are proposed to compress videos for efficient storage, *i.e.*, compress one frame by reusing contents from another frame (termed *reference frame*) and only store the change. According to the choice of reference frame, frames are categorized into three types: I-frame (Intra-coded frame), P-frame (Predictive frame), and B-frame (Bi-predictive frame). I-frame has no reference frame and is directly stored in image format, while P- and B-frame take reference frames forwardly and bi-directionally, respectively, and store the change in the format of the motion vector and residual: A codec first divides a frame into macroblocks of size such

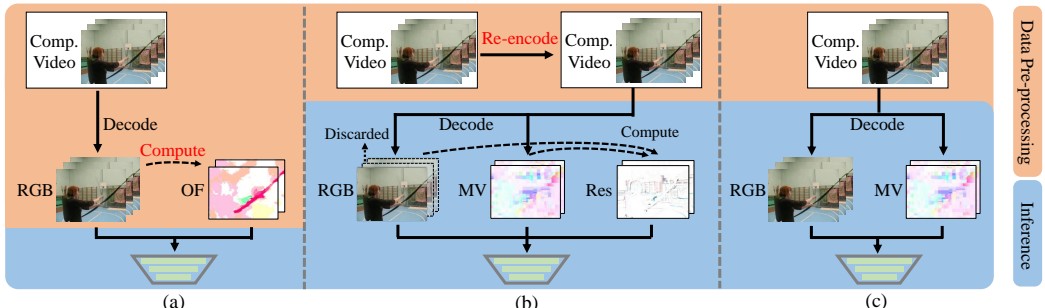

Figure 2: **Usage of compressed videos (Comp. Video)** for SSVRL: (a) Traditional methods decode videos off-the-fly; (b) IMRNet re-encodes videos, followed by the additional residuals computation with RGB frames being discarded afterward; (c) MVCGC decodes RGB frames and motion vectors on-the-fly. "RGB", "MV", "OF", and "Res" denote RGB frames, motion vectors, optical flows, and residuals, respectively. Procedures colored with red are computation expensive, and their backgrounds, which are colored with orange or blue, indicate the happening place of them, *i.e.*, videos in the orange background are stored on the disk while those in the blue are decoded in cache memory.

as 16x16 and then searches the most similar image patch in the reference frame for each of these macroblocks. Lastly, the motion vector is represented by the displacements between macroblocks and the searched image patches, and the residual is computed as the difference between the target frame and its reference frame warped by the motion vector.

**Discussion**     As shown in Figure 2, existing SSVRL methods can be categorized into three groups according to how they exploit videos for SSVRL. (1) Traditional methods off-the-fly decode all RGB frames from compressed videos and compute optical flows with RGB frames, which is storage- and computation-intensive. (2) The latest IMRNet [Yu *et al.*, 2021] decodes compressed videos by leveraging the process in CoViAR [Wu *et al.*, 2018], which only supports the outdated MPEG-4 Part 2 codec [Sikora, 1997] and lacks generalizability. In this way, videos with modern codecs (*e.g.*, H.264/AVC [Wiegand *et al.*, 2003] and HEVC [Sullivan *et al.*, 2012]) in current application scenarios need to be re-encoded and re-stored, downgrading the computational efficiency. IMRNet uses residuals of P-frames as a third view: since residuals can not be extracted in practice, it is post-computed by two decoded views, *i.e.*, RGB frames and motion vectors, which introduces extra computation requirements. Moreover, in IMRNet, RGB frames are decoded only for computing residuals, while being discarded afterward, hindering the data usage effectiveness. (3) Our proposed MVCGC can decode RGB frames and motion vectors directly from various video codecs without the aforementioned costly off-the-fly frame extracting, video re-encoding, or residual computing, and is thus high-efficiency and application-flexibility.

### 3.2    Motion Vector based Cross Guidance Contrastive Learning

The goal of SSVRL is to train an encoder that can embed video samples effectively for various downstream tasks (*e.g.*, action recognition and retrieval). Towards this end, our method is based on contrastive learning with InfoNCE loss [Oord *et al.*, 2018], as shown in Figure 3. Next we start by reviewing a base single-view model with InfoNCE loss, and then extend the base model using cross guidance contrastive learning as our MVCGC.

**Base Single-View Model with InfoNCE Loss**    Formally, let $\mathcal{D} = \{x_1, x_2, \cdots, x_N\}$ denotes a dataset of $N$ compressed video samples, where each clip $x_i$ ($i = 1, 2, \cdots, N$) consists of $T$ frames. In the single-view scenario, these clips are in the same view, *i.e.*, $\mathcal{D}$ is a dataset of only RGB clips or motion vectors. The objective of learning an effective encoding function $f(\cdot)$ for the single stream is thus achieved by discriminating one positive sample of each video clip from its negatives with InfoNCE loss [Oord *et al.*, 2018].

Let $t(\cdot; \theta)$ be an augmentation function applied to $\mathcal{D}$, where $\theta$ is sampled from a set of transformations $\Theta$. For each clip $x_i \in \mathcal{D}$ ($i = 1, 2, \cdots, N$), we denote the encoded feature vector of $x_i$ as $z_i = f(x_i)$, the positive sample set as $\mathcal{P}_i = \{f(t(x_i; \theta)) | \theta \sim \Theta\}$, and the negative set as $\mathcal{N}_i = \{f(t(x_j; \theta)) | \forall j \neq$

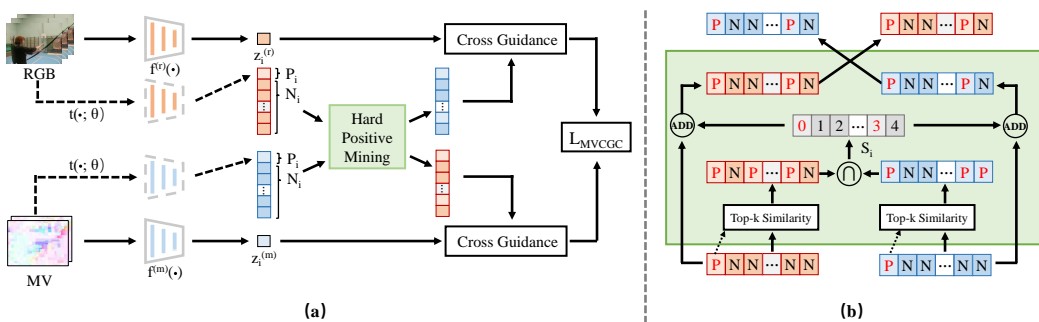

**Figure 3:** Overview of the proposed MVCGC architecture. (a) From a raw RGB clip and its corresponding motion vector clip, MVCGC computes the anchor and the positive features, while other samples in the dataset are used to construct the negative set. (b) The complete design of the hard positive mining stage.

$i, \theta \sim \Theta\}$, where $|\mathcal{P}_i| = 1$ and $|\mathcal{N}_i| = N - 1$. The InfoNCE loss is then defined as:

$$\mathcal{L}_{\text{InfoNCE}} = \sum_i -\log \frac{\exp\left(z_i \cdot z_p/\tau\right)}{\exp\left(z_i \cdot z_p/\tau\right) + \sum_{z_n \in \mathcal{N}_i} \exp\left(z_i \cdot z_n/\tau\right)}, \tag{1}$$

where $z_p \in \mathcal{P}_i$ is the only positive sample of $x_i$, $z_i \cdot z_p$ (or $z_i \cdot z_n$) refers to the dot product between two vectors, and $\tau > 0$ is a temperature parameter. This instance discrimination loss forces the base model to learn a higher similarity score between a given sample $z_i$ and its augmented/target sample $z_p$ while a lower one between $z_i$ and *other* instances $z_n \in \mathcal{N}_i$.

**Cross Guidance Contrastive Learning with Multiple Views**   While the InfoNCE loss can learn discriminative representations in a single view well, it faces two key challenges when being extended to multiple views: (1) For multiple sources of information (*i.e.*, RGB frames, motion vectors, and optical flows), simply applying an InfoNCE loss for each stream ignores the corresponding mutual information across these streams. (2) It aims only to discriminate the augmented sample from all other instances, with the assumption that the target sample is semantically different from all the others. This is counter-intuitive, since different video clips may also have similar semantic content, *i.e.*, hard positives are neglected by InfoNCE. We thus propose MVCGC to address these two problems.

Concretely, MVCGC decodes an RGB stream (**RGB**) $r_i \in \mathbb{R}^{T \times H \times W \times 3}$ and a motion vector stream (**MV**) $m_i \in \mathbb{R}^{T \times H \times W \times 2}$ as two input views for each video clip $x_i \in \mathcal{D}$ ($i = 1, 2, \cdots, N$). MVCGC aims to learn two encoding functions $f^{(r)}(\cdot), f^{(m)}(\cdot)$ for the two streams, respectively. To model the interaction between two streams, we extend the single-view InfoNCE loss in Eq. (1) by choosing positive and negative samples from the other view for each stream. Specifically, given the query of an RGB clip $z_i^{(r)}$, the objective is to compare its similarity among the features of motion vector clips, *i.e.*, emitting higher similarity between the positive sample $z_p^{(m)}, p \in \mathcal{P}_i$ than with those of other negative instances $z_n^{(m)}, n \in \mathcal{N}_i$. The positive sets (*i.e.*, $\mathcal{P}_i^{(r)}$ and $\mathcal{P}_i^{(m)}$) of each $r_i$ or $m_i$ are also expanded for the hard positive mining by adding the top-k similar clips:

$$\mathcal{P}_i^{(r)} = \{f^{(m)}(t(m_j; \theta)) | j \in \mathcal{S}_i, \theta \sim \Theta\}, \tag{2}$$

$$\mathcal{P}_i^{(m)} = \{f^{(r)}(t(\ r_j; \theta)) | j \in \mathcal{S}_i, \theta \sim \Theta\}, \tag{3}$$

where $\mathcal{S}_i = \{j | j \in (\text{top-k}(z_i^{(r)} \cdot z_j^{(r)}) \cap \text{top-k}(z_i^{(m)} \cdot z_j^{(m)})), j = (1, 2, \cdots, N)\}$ denotes the index set of positive samples, $z_i^{(r)} = f^{(r)}(r_i)$ and $z_i^{(m)} = f^{(m)}(m_i)$. Our learning objective is thus to use the features in the two views as the cross guidance of each other:

$$\mathcal{L}_{\text{MVCGC}}^i = -\log \frac{\sum_{z_p^{(m)}} \exp(z_i^{(r)} \cdot z_p^{(m)}/\tau)}{\sum_{z_p^{(m)}} \exp(z_i^{(r)} \cdot z_p^{(m)}/\tau) + \sum_{z_n^{(m)}} \exp(z_i^{(r)} \cdot z_n^{(m)}/\tau)}$$
$$-\log \frac{\sum_{z_p^{(r)}} \exp(z_i^{(m)} \cdot z_p^{(r)}/\tau)}{\sum_{z_p^{(r)}} \exp(z_i^{(m)} \cdot z_p^{(r)}/\tau) + \sum_{z_n^{(r)}} \exp(z_i^{(m)} \cdot z_n^{(r)}/\tau)}, \tag{4}$$

where $z_p^{(m)} \in \mathcal{P}_i^{(r)}$, $z_n^{(m)} \in \mathcal{N}_i^{(r)}$, $z_p^{(r)} \in \mathcal{P}_i^{(m)}$, and $z_n^{(r)} \in \mathcal{N}_i^{(m)}$. The final loss is computed over all training samples:

$$\mathcal{L}_{\text{MVCGC}} = \sum_i \mathcal{L}_{\text{MVCGC}}^i. \tag{5}$$

**Discussion**    Note that our MVCGC is different from other methods with discriminative InfoNCE loss in two aspects. First, differing from CMC [Tian *et al.*, 2020] and IIC [Tao *et al.*, 2020] which only mine a single positive and regard the hard positives as negative samples, MVCGC incorporates the learning from hard positives. Second, the co-training in CoCLR [Han *et al.*, 2020b] fails to learn the correspondence between two views (*i.e.*, there is no information exchange in the pre-learning stage), but our MVCGC leverages cross guidance to capture mutual information between two streams. We provide the algorithm in the supplementary material.

### 3.3 MVCGC Algorithm

The detailed procedure of our MVCGC is summarized in Algorithm 1.

---

**Algorithm 1** Motion Vector based Cross Guidance Contrastive Learning (MVCGC)

---

**Require:** Compressed video dataset $\mathcal{D}$.
  An RGB clip $r_i \in \mathbb{R}^{T \times H \times W \times 3}$ and a motion vector clip $m_i \in \mathbb{R}^{T \times H \times W \times 2}$ for $x_i \in \mathcal{D}$
  $(i = 1, 2, \cdots, N)$.
  Encoders $f^{(r)}(\cdot)$, $f^{(m)}(\cdot)$ for RGB and motion vector streams, respectively.
  The augmentation set $t(\cdot; \theta), \theta \sim \Theta$.
  Temperature $\tau$, top-k hard positive mining parameter $k$.
1: **for** $epoch = 1$ **to** #epochs: **do**
2:   Compute $z_i^{(r)}, z_i^{(m)}, z_j^{(r)}, z_j^{(m)}, \forall i \in [1, N], j \neq i$;
3:   Compute $\mathcal{S}_i$;
4:   Obtain $\mathcal{P}_i^{(r)}, \mathcal{P}_i^{(m)}, \mathcal{N}_i^{(r)}, \mathcal{N}_i^{(m)}$;
5:   Compute cross-entropy loss $\mathcal{L}_{\text{MVCGC}}$;
6:   Update model parameters;
7: **end for**
8: **return** Optimized encoders $f^{(r)}(\cdot)$ and $f^{(m)}(\cdot)$.

---

## 4 Experiments

### 4.1 Datasets

In this paper, we use UCF101 [Soomro *et al.*, 2012] and Kinetics-400 (K400) [Kay *et al.*, 2017] for self-supervised pre-training. UCF101 contains 13,320 videos with 101 action classes and has three standard training/test splits. While K400 is a larger dataset consisting of 400 human action classes and has 230k/20k clips for training/validation, respectively. In the pre-training stage, compressed videos in the first training set of UCF101 and the training split of K400 are used without labels. Following the common practice [Han *et al.*, 2020b], we benchmark downstream evaluation tasks on the first test set of UCF101, and the test split 1 of HMDB51 [Kuehne *et al.*, 2011], a relatively small action dataset containing 6,766 videos with 51 categories.

### 4.2 Implementation details

We follow the state-of-the-art method [Han *et al.*, 2020b] for adopting S3D [Xie *et al.*, 2018] architecture as the backbone feature extractor for all experiments and apply the momentum-updated history queue as in MoCo [He *et al.*, 2020] for the framework of contrastive learning to leverage a larger number of negative samples. Datasets used contain videos with various codecs, which are directly decoded into RGB frames and motion vectors. We implement MVCGC based on "pyav", a Pythonic binding for the FFmpeg libraries used for decoding RGB frames and motion vectors on-the-fly. A non-linear projection head is attached above each encoder during the pre-training stage and is removed for downstream evaluations.

Table 1: Results of processing speeds, storages and accuracies on UCF101. The underline represents the second-best result.

| Method | IIC [Tao et al., 2020] | MemDPC [Han et al., 2020a] | CoCLR [Han et al., 2020b] | IMRNet [Yu et al., 2021] | MVCGC (ours) |
|---|---|---|---|---|---|
| Pre-processing Time (s) ↓ | 12.7 | 12.7 | 13.2 | 0.128 | **0.008** |
| Inference Time (ms) ↓ | **69.3** | 253.1 | 248.7 | 227.2 | 70.3 |
| Total Time (s) ↓ | 12.8 | 13.0 | 13.4 | 0.355 | **0.078** |
| Storage (GB)↓ | 38.0 | 38.0 | 35.4 | 6.1 | **1.9** |
| Top-1 Acc (%)↑ | 72.7 | 84.3 | 87.3 | 76.8 | **87.4** |

Our MVCGC proceeds in two stages: the initialization stage and the cross guidance stage. In the first stage, the encoder of each view (*i.e.*, **RGB** or **MV**) is initialized with Eq. (1) independently. After initialization, these two encoders are cross-trained together to minimize the loss in Eq. (5). To cache a large number of features, we adopt a momentum-updated history queue as in MoCo [He *et al.*, 2020], which is used in both two pre-training stages. For the pre-training on UCF101, temperature $\tau = 0.07$, momentum $m = 0.999$ and queue size $2048$ are used, while queue size is set to $16384$ on K400. When pre-training on UCF101, the initialization stage lasts 300 epochs for each stream, and we then continually train the cross guidance for another 200 epochs. On K400, we train 200 epochs for each stream in the initialization stage and 50 epochs for cross guidance contrastive learning. 100 and 500 epochs are used for linear and fully fine-tuning, respectively. We use the Adam optimizer with a 1e-4 learning rate and 1e-5 weight decay for pre-training and the SGD optimizer with a 1e-1 learning rate and 1e-3 weight decay for fine-tuning. The learning rate is decayed down by 1/10 twice when the validation loss plateaus. The hyper-parameter $k$ in MVCGC is set as 5 according to the ablation study. All experiments are trained on 4 TiTan RTX GPUs, with a batch size of 32 samples per GPU.

We make evaluation on two downstream tasks: action classification and video retrieval. In the action classification task, two evaluation settings are used: (1) *linear probing*, where parameters of the learned encoder are frozen and only a single linear layer is fine-tuned; (2) *fully fine-tuning*, where the encoder is fine-tuned together. At the inference stage in either of these two settings, we follow the same procedure as in the previous work [Han *et al.*, 2020b]: for each video, we spatially apply ten-crops and temporally take clips with moving windows, and then average the predicted probabilities. In the video retrieval task, we adopt widely-used evaluation metrics [Han *et al.*, 2020b; Tao *et al.*, 2020], *i.e.*, we leverage the extracted features for nearest-neighbour (NN) retrieval without fine-tuning. More details can be found in the supplementary material.

### 4.3 Efficiency Results

Table 1 presents the efficiency results on UCF101 compared with four state-of-the-art self-supervised methods: IIC [Tao *et al.*, 2020], MemDPC [Han *et al.*, 2020a], CoCLR [Han *et al.*, 2020b], and IMRNet [Yu *et al.*, 2021]. We use per-video processing time, storage spaces, and top-1 accuracies as evaluation metrics under the fully fine-tuning setting. All methods are measured in exactly the same environment: Intel Xeon 5118 CPUs and a Titan RTX GPU. The total processing time consists of two parts: data pre-processing and inference time. In the pre-processing stage, IIC, MemDPC, and CoCLR decode RGB frames and calculate optical flows with the TV-L1 algorithm, while IMRNet re-encodes the raw compressed videos. We can observe that: (1) our MVCGC has the fastest speed and reduces the storage budgets by 95% and 70% compared with traditional methods and IMRNet, respectively. (2) For inference, the data is loaded from disks and is used for network forwarding. Although the time costs may differ because of the backbone architectures used by different methods, MVCGC still gets a comparable result to IIC and is faster than other three approaches. Nevertheless, the majority of the total process is dominated by data pre-processing, where MVCGC significantly outperforms competitors. (3) Among fully fine-tuning methods, MVCGC outperforms those using either residuals or optical flows, showing the effectiveness of our proposed algorithm.

### 4.4 Ablation Study

We first show how our proposed MVCGC benefits the learned representations of motion vectors in Table 2, where the top-1 accuracies of action classification and retrieval performance are used as evaluation metrics since their evaluation is fast. We can make the following four observations: (1) The upper part of the table shows the results of different views after initialization, where there is clearly a

Table 2: Evaluation of motion vectors (MV) and optical flows (OF) on downstream action classification and retrieval on UCF101.

| Method | Pre-training | Evaluation | Linear Probe | Retrieval |
|--------|--------------|------------|--------------|-----------|
| Init. | MV | MV | 65.0 | 39.9 |
| Init. | OF | OF | 66.8 | 45.2 |
| Init. | $OF_{Low}$ | $OF_{Low}$ | 51.3 | 30.4 |
| Init. | $MV_{CoViAR}$ | $MV_{CoViAR}$ | 63.4 | 39.2 |
| MVCGC | RGB+OF | OF | 73.1 (+6.3) | 60.6 (+15.4) |
| MVCGC | RGB+MV | MV | 73.8 (+8.8) | 60.8 (+20.9) |

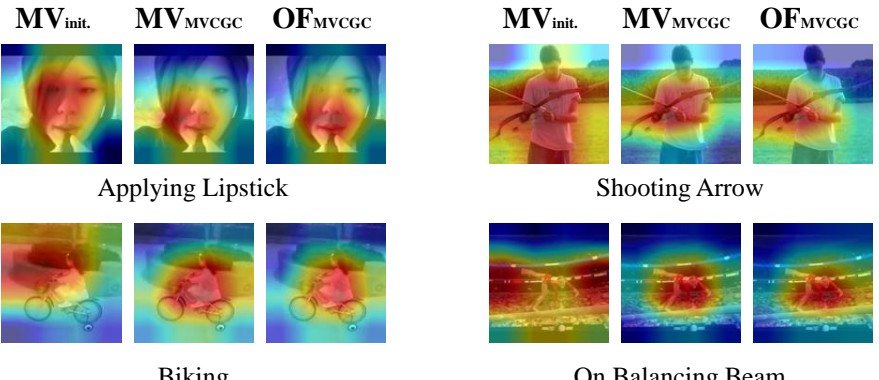

**MV**_init.   **MV**_MVCGC   **OF**_MVCGC          **MV**_init.   **MV**_MVCGC   **OF**_MVCGC

Applying Lipstick                    Shooting Arrow

Biking                         On Balancing Beam

Figure 4: Visualization of the last feature maps from the models linear probed on UCF101. Each row shows the attention visualization of motion vectors before and after MVCGC pre-training and the final result of optical flows, respectively. Attention map is generated with 32-frames clip inputs and applied to the middle frame in the video clips.

performance gap between the motion vector and the optical flow. To verify whether this gap originates from the lower resolution of the motion vector (*i.e.*, same value in each macroblock), we also compare with the lower resolution of the optical flow (*i.e.*, $OF_{Low}$). We find that by only reducing the resolution of the raw optical flow to a half (*i.e.*, same value in each $2 \times 2$ macroblock), it performs even worse than the motion vector. This comparison demonstrates that the motion vector, although having lower resolutions than the optical flow, is also able to well represent the action motions. (2) Since motion vectors may appear differently based on codec settings (*i.e.*, different codecs and coding qualities), we also conduct experiment with the motion vectors extracted by the previous procedure [Yu *et al.*, 2021; Wu *et al.*, 2018] (*i.e.*, $MV_{CoViAR}$), where they are all decoded from P-frames. We find that our MVCGC is robust to motion vectors under different encoding configurations. (3) The motion vector view (or the optical flow view) can significantly benefit from the RGB view by our MVCGC, validating the effectiveness of our proposed cross guidance pre-training used in our MVCGC. (4) The performance of our MVCGC using motion vector + RGB is comparable to (and even slightly better than) that of our MVCGC using optical flow + RGB. This finding is also supported by the visualization results shown in Figure 4.

We then demonstrate the advantages of our proposed MVCGC over the closely-related method CoCLR [Han *et al.*, 2020b] and also our choices for the hyper-parameters of MVCGC in Table 3, where we focus on the performance of the RGB network during evaluation. First, from the upper part of Table 3, we find that our MVCGC indeed helps the RGB network to learn more discriminative representations than CoCLR, when the extra information of the motion vector view (or the optical flow view) is leveraged. This shows the effectiveness of our cross guidance contrastive learning. Second, according to the lower part of Table 3, we select $k = 5$ as the optimal setting among $k = 1/5/10/20$. Particularly, MVCGC with 'w/o init.' means that there is no initialization stage before the cross guidance stage, which performs the worse. Moreover, MVCGC with 'union' means that the index set $\mathcal{S}_i$ is generated by the union of positive candidates in two streams, which performs worse than the intersection used in our MVCGC.

Table 3: Evaluation of different pre-training settings (*i.e.*, algorithms and hyper-parameters) on downstream action classification and video retrieval on UCF101. All results are evaluated using only the RGB view. The underline represents the second-best result.

| Method | Pre-training | Evaluation | Linear Probe | Retrieval |
|---|---|---|---|---|
| Init. | RGB | RGB | 52.3 | 33.1 |
| CoCLR | RGB+OF | RGB | 70.2(+17.9) | 51.8(+18.7) |
| CoCLR | RGB+MV | RGB | 67.4(+15.1) | 53.8(+20.7) |
| MVCGC | RGB+OF | RGB | 70.6(+18.3) | 58.6(+25.5) |
| **MVCGC$_{(k=5)}$** | RGB+MV | RGB | **73.1(+20.8)** | 60.8(+27.7) |
| MVCGC$_{(k=5, w/o init.)}$ | RGB+MV | RGB | 65.0 | 48.4 |
| MVCGC$_{(k=1)}$ | RGB+MV | RGB | 71.5 | 60.0 |
| MVCGC$_{(k=10)}$ | RGB+MV | RGB | 72.7 | **61.4** |
| MVCGC$_{(k=20)}$ | RGB+MV | RGB | 71.5 | 59.7 |
| MVCGC$_{(k=5, union)}$ | RGB+MV | RGB | 71.3 | 57.7 |

Table 4: The full table of comparison with state-of-the-art approaches on linear probing (Lin.) and fully fine-tuning (Full.) on UCF101 and HMDB51 benchmarks. Top-1 accuracies are reported. "Dataset" denotes the pre-training dataset and "Duration" is the total length of videos. Methods marked with * use multiple views for pre-training. 'Test View(s)' denotes views used in evaluation, *i.e.*, RGB, motion vector(MV), optical flow(OF), residual(Res), Audio, and Text.

| Method | Test View(s) | Arch. | Res. | Dataset(Duration) | UCF Lin. | UCF Full. | HMDB Lin. | HMDB Full. |
|---|---|---|---|---|---|---|---|---|
| VCP [Luo *et al.*, 2020] | RGB | C3D | 16*112 | UCF(1d) | – | 68.5 | – | 32.5 |
| PRP [Yao *et al.*, 2020] | RGB | R(2+1)D-18 | 16*112 | UCF(1d) | – | 72.1 | – | 35.0 |
| Pace [Wang *et al.*, 2020] | RGB | R(2+1)D-18 | 16*112 | UCF(1d) | – | 75.9 | – | 35.9 |
| DSM* [Wang *et al.*, 2021] | RGB | C3D | 16*112 | UCF(1d) | – | 70.3 | – | 40.5 |
| CMC* [Tian *et al.*, 2020] | RGB | CaffeNet | 25*256 | UCF(1d) | – | 59.1 | – | 26.7 |
| CoCLR* [Han *et al.*, 2020b] | RGB | S3D | 32*128 | UCF(1d) | 70.2 | 81.4 | 39.1 | 52.1 |
| **MVCGC (ours)*** | RGB | S3D | 32*128 | UCF(1d) | **73.1** | **82.0** | **41.1** | **58.4** |
| IIC* [Tao *et al.*, 2020] | RGB+OF | R3D-18 | 16*112 | UCF(1d) | – | 72.7 | – | 36.8 |
| MemDPC* [Han *et al.*, 2020a] | RGB+OF | R2D3D-34 | 40*128 | UCF(1d) | – | 84.3 | – | – |
| CoCLR* [Han *et al.*, 2020b] | RGB+OF | S3D | 32*128 | UCF(1d) | 72.1 | 87.3 | 40.2 | 58.7 |
| **MVCGC (ours)*** | RGB+MV | S3D | 32*128 | UCF(1d) | **77.2** | **87.4** | **41.0** | **59.7** |
| CCL [Kong *et al.*, 2020] | RGB | R(2+1)D-18 | 8*112 | K400(28d) | 52.1 | 69.4 | 27.8 | 37.8 |
| DPC [Han *et al.*, 2019] | RGB | R2D3D-34 | 40*224 | K400(28d) | – | 75.7 | – | 35.7 |
| Pace [Wang *et al.*, 2020] | RGB | R(2+1)D-18 | 16*112 | K400(28d) | – | 77.1 | – | 36.6 |
| SpeedNet [Benaim *et al.*, 2020] | RGB | S3D-G | 16*224 | K400(28d) | – | 81.1 | – | 48.8 |
| DSM* [Wang *et al.*, 2021] | RGB | R3D-34 | 16*224 | K400(28d) | – | 78.2 | – | 52.8 |
| VIE [Zhuang *et al.*, 2020] | RGB | SlowFast-18 | 16*112 | K400(28d) | – | 80.4 | – | 52.5 |
| CoCLR* [Han *et al.*, 2020b] | RGB | S3D | 32*128 | K400(28d) | 74.5 | 87.9 | 46.1 | 54.6 |
| **MVCGC (ours)*** | RGB | S3D | 32*128 | K400(28d) | **75.4** | **88.3** | **49.7** | **61.4** |
| AVTS* [Korbar *et al.*, 2018] | RGB+Audio | R(2+1)D-18 | 25*224 | K400(28d) | – | 86.2 | – | 52.3 |
| ELO* [Piergiovanni *et al.*, 2020] | RGB+Audio+OF | R(2+1)D-50 | 32*224 | YouTube800M(1.9y) | – | 93.8 | 64.5 | 67.4 |
| XDC* [Alwassel *et al.*, 2020] | RGB+Audio | R(2+1)D-18 | 32*224 | K400(28d) | – | 86.8 | – | 52.6 |
| GDT* [Patrick *et al.*, 2020] | RGB+Audio | R(2+1)D-18 | 32*224 | K400(28d) | – | 89.3 | – | 60.0 |
| CBT* [Sun *et al.*, 2019] | RGB+Text | S3D | 16*112 | K400(28d) | 54.0 | 79.5 | 29.5 | 44.6 |
| MIL-NCE* [Miech *et al.*, 2020] | RGB+Text | S3D | 32*200 | Howto100M(15y) | 82.7 | 91.3 | 53.1 | 61.0 |
| IMRNet* [Yu *et al.*, 2021] | RGB+MV+Res | R3D-50 | 30*224 | K400(28d) | – | 77.3 | – | 47.5 |
| MemDPC* [Han *et al.*, 2020a] | RGB+OF | R2D3D-34 | 40*224 | K400(28d) | 54.1 | 86.1 | 30.5 | 54.5 |
| CoCLR* [Han *et al.*, 2020b] | RGB+OF | S3D | 32*128 | K400(28d) | 77.8 | 90.6 | 52.4 | 62.9 |
| **MVCGC (ours)*** | RGB+MV | S3D | 32*128 | K400(28d) | **78.0** | **90.8** | **53.0** | **63.4** |

## 4.5 Comparison with State-of-the-Art

In this section, we compare our MVCGC with the state-of-the-art self-supervised SSVRL approaches on the action classification and video retrieval tasks. For the action classification task, we provide the comparative results in Table 4, by pre-training on UCF101 and Kinetics-400 and then evaluating with either linear probing or fully fine-tuning on UCF101 and HMDB51. We list recent approaches evaluated on the same benchmark, trying to compare with them as fairly as we can, although variations are unavoidable in terms of architecture, training data, and resolution. The 'Test View(s)' column refers to the view(s) used for test/evaluation, *e.g.*, the result of MVCGC in 'R+MV' views is obtained by averaging the predictions from RGB and motion vector networks. We can observe that: (1)

Table 5: Comparison with state-of-the-art methods in nearest-neighbour video retrieval on UCF101 and HMDB51. Videos in the test set are used to retrieve the videos in the training set, and R@k is reported with $k \in \{1, 5, 10, 20, 50\}$. All compared methods are pre-trained on UCF101 except SpeedNet pre-trained on larger Kinetics-400. Methods marked with $^*$ use audio for pre-training.

| Method | UCF | | | | | HMDB | | | | |
|---|---|---|---|---|---|---|---|---|---|---|
| | R@1 | R@5 | R@10 | R@20 | R@50 | R@1 | R@5 | R@10 | R@20 | R@50 |
| **Retrieval with RGB only:** | | | | | | | | | | |
| VCP [Luo *et al.*, 2020] | 18.6 | 33.6 | 42.5 | 53.5 | 68.1 | 7.6 | 24.4 | 36.3 | 53.6 | 76.4 |
| SpeedNet [Benaim *et al.*, 2020] | 13.0 | 28.1 | 37.5 | 49.5 | 65.0 | – | – | – | – | – |
| DSM [Wang *et al.*, 2021] | 16.8 | 33.4 | 43.4 | 54.6 | 70.7 | 8.2 | 25.9 | 38.1 | 52.0 | 75.0 |
| PRP [Yao *et al.*, 2020] | 23.2 | 38.1 | 46.0 | 55.7 | 68.4 | 10.5 | 27.2 | 40.4 | 56.2 | 75.9 |
| MemDPC [Han *et al.*, 2020a] | 20.2 | 40.4 | 52.4 | 64.7 | – | 7.7 | 25.7 | 40.6 | 57.7 | – |
| Pace [Wang *et al.*, 2020] | 31.9 | 49.7 | 59.2 | 68.9 | 80.2 | 12.5 | 32.2 | 45.4 | 61.0 | 80.7 |
| CCL [Kong *et al.*, 2020] | 32.7 | 42.5 | 50.8 | 61.2 | 68.9 | – | – | – | – | – |
| CoCLR [Han *et al.*, 2020b] | 53.3 | 69.4 | 76.6 | 82.0 | – | 23.2 | 43.2 | 53.5 | 65.5 | – |
| GDT$^*$ [Patrick *et al.*, 2020] | 57.4 | 73.4 | 80.8 | 88.1 | 92.9 | 25.4 | 51.4 | 63.9 | 75.0 | 87.8 |
| **MVCGC (ours)** | **60.8** | **74.1** | **79.8** | **85.8** | **92.6** | 24.1 | 49.7 | 61.3 | 73.3 | 87.5 |
| **Retrieval with two views:** | | | | | | | | | | |
| IIC [Tao *et al.*, 2020] | 42.4 | 60.9 | 69.2 | 77.1 | 86.5 | 19.7 | 42.9 | 57.1 | 70.6 | 85.9 |
| CoCLR [Han *et al.*, 2020b] | 55.9 | 70.8 | 76.9 | 82.5 | – | **26.1** | 45.8 | 57.9 | 69.7 | – |
| **MVCGC (ours)** | **66.1** | **79.1** | **84.0** | **89.0** | **93.7** | 25.6 | **49.2** | **60.7** | **73.3** | **86.5** |

Our MVCGC outperforms most of latest methods under both linear probing and fully fine-tuning settings on both benchmarks. (2) Our MVCGC using motion vectors from compressed videos clearly beats CMC [Tian *et al.*, 2020] and CoCLR [Han *et al.*, 2020b] which exploit optical flows for pre-training. (3) Compared to IMRNet [Yu *et al.*, 2021] that also utilizes compressed videos for self-supervised learning, our method achieves significantly better results, demonstrating both efficiency and effectiveness of MVCGC. (4) Our MVCGC outperforms its competitors under the pre-training on Kinetics-400 and fine-tuning on UCF101/HMDB51 setting, indicating the robustness of MVCGC to process motion vectors that are obtained with different codecs. (5) It can be observed that we outperform some approaches that exploit the correspondence of visual information with text or audio. This is impressive since we only focus on visual representations and only use visual-related views (*i.e.*, RGB, motion vectors, optical flows).

In Table 5, we evaluate our MVCGC in video retrieval on both UCF101 and HMDB51 We can see that our MVCGC significantly outperforms all compared methods in terms of all five metrics on UCF101 and four of the five metrics on HMDB51, achieving new state-of-the-art. Particularly, our obtained retrieval results by retrieval with only motion vectors are even higher than those of the latest competitor CoCLR [Han *et al.*, 2020b] which uses optical flows for retrieval, indicating the effectiveness of our MVCGC. Furthermore, we obtain comparable results with methods using extra audio modalities [Patrick *et al.*, 2020], indicating the effectiveness of MVCGC.

## 5 Conclusion

We have proposed a novel self-supervised video representation learning method named Motion Vector based Cross Guidance Contrastive Learning (MVCGC). By introducing on-the-fly decoded motion vectors and cross guidance contrastive learning, our proposed MVCGC has for the first time leveraged contrastive loss in compressed video self-supervised learning. Extensive experiments are carried out to validate the efficiency and effectiveness of MVCGC. Importantly, our approach achieves new state-of-the-art on two downstream tasks across different benchmarks.

## Acknowledgments and Disclosure of Funding

This work was supported in part by National Natural Science Foundation of China (61976220 and 61832017), Beijing Outstanding Young Scientist Program (BJJWZYJH012019100020098), and Open Project Program Foundation of Key Laboratory of Opto-Electronics Information Processing, Chinese Academy of Sciences (OEIP-O-202006). Ping Luo was supported by the General Research Fund of Hong Kong No.27208720.

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
