# Supplementary Material for Compressed Video Contrastive Learning

**Yuqi Huo**[2,3,•]  **Mingyu Ding**[4,•]  **Haoyu Lu**[1,2]  **Nanyi Fei**[2,3]
**Zhiwu Lu**[1,2,*]  **Ji-Rong Wen**[1,2]  **Ping Luo**[4]

[1]Gaoling School of Artificial Intelligence, Renmin University of China, Beijing, China
[2]Beijing Key Laboratory of Big Data Management and Analysis Methods
[3]School of Information, Renmin University of China, Beijing, China
[4]The University of Hong Kong, Pokfulam, Hong Kong, China
{bnhony, luzhiwu}@ruc.edu.cn
• Equal contribution       * Corresponding author

## A   Implementation Details

Datasets used contain videos with various codecs, which are directly decoded into RGB frames and motion vectors. We implement MVCGC based on "pyav", a Pythonic binding for the FFmpeg libraries used for decoding RGB frames and motion vectors on-the-fly. We sample a 32-frame clip from each decoded video with a spatial resolution of $128 \times 128$ pixels. We apply clip-wise random crops, horizontal flips, Gaussian blur, graying, and color jittering in the spatial dimension while random sampling a continuous 32-frame clip in the temporal dimension for data augmentation. Motion vectors are processed as follows: those of I-frames are directly set as all-zero matrices; and the forward motion vectors in B-frames are used, whose information resembles those in P-frames. A third zero channel is stacked to the two-channel motion vector, and motions exceeding 15 pixels are truncated. The final values are projected to [0, 255] as RGB images, and are only spatially augmented (*i.e.*, crops and flips). A non-linear projection head is attached above each encoder during the pre-training stage and is removed for downstream evaluations.

## B   Architecture

We use the S3D architecture [Xie *et al.*, 2018] for all experiments following the previous work [Han *et al.*, 2020]. At the pre-training stage, S3D is followed by a non-linear projection head, which consists of two fully-connected (FC) layers and is removed when evaluating downstream tasks. The detailed dimensions are shown in Table 1.

Table 1: The encoder architecture of both RGB and motion vector streams at the pre-training stage. "FC-1024" and "FC-128" denote the output dimension of each fully-connected layer, respectively.

| Stage | Detail | Output size: T×H(W)×C |
|---|---|---|
| Backbone | followed by average pooling | $1 \times 1^2 \times 1024$ |
| Projection | FC-1024→ReLU→FC-128 | $1 \times 1^2 \times 128$ |

Once pre-trained, we replace the non-linear projection head with a single linear layer for classification tasks. The detailed dimensions are illustrated in Table 2.

Table 2: The classifier architecture for evaluating the representations on the action classification tasks. Note that "FC-num_class" denotes the output dimension of the fully-connected layer (*i.e.*, the number of action classes).

| Stage | Detail | Output size: T×H(W)×C |
|---|---|---|
| S3D | followed by average pooling | $1 \times 1^2 \times 1024$ |
| Linear layer | one layer: FC-num_class | $1 \times 1^2 \times$ num_class |

**Query**                 **Top3 Nearest Neighbours**

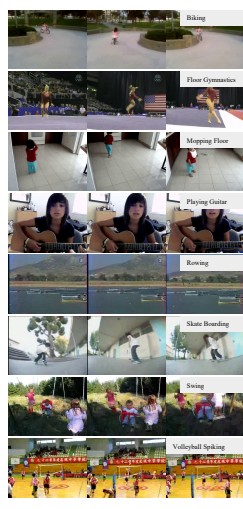 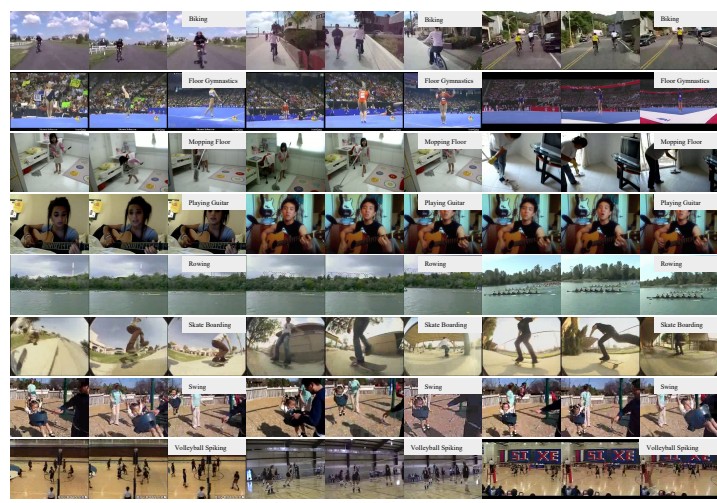

Figure 1: Nearest neighbour retrieval results with MVCGC representations. The left parts are the query videos from the UCF101 testing set, and the right parts show the Top-3 nearest neighbours from the UCF101 training set. The action label for each video is shown in the upper right corner.

## C  Qualitative Results for Video Retrieval

Here we demonstrate the qualitative results for video retrieval. Figure 1 visualizes query video clips and their Top-3 Nearest Neighbors from the UCF101 training set using the concatenation of MVCGC two stream embeddings. As shown in the figure, the representations learned by our MVCGC are able to retrieve videos belonging to the same semantic categories.