# OpenReview forum: "Compressed Video Contrastive Learning"
_NeurIPS.cc/2021/Conference — NeurIPS 2021 Poster_

### Official Review · Reviewer_M3h9 · 2021-07-06

**Rating:** 6
**Confidence:** 4

**Summary:**

The paper tackles the problem of self-supervised video representation learning. The authors propose a compressed video based self-supervised learning approach, where the model is forced to learn matching representations between raw RGB clip and its corresponding motion vector clip. This is achieved using a relatively standard contrastive learning formulation. The authors then demonstrate that their model achieves state-of-the-art performance on a few downstream tasks (i.e., action recognition, and video retrieval) while also being significantly more efficient (particularly in terms of the video processing time).

**Limitations And Societal Impact:**

Limitations were not discussed.

**Main Review:**

Strengths:
- The paper is written clearly, and it addresses a well motivated and an important research problem.
- A nice idea of using compressed video for self-supervised feature learning.
- The proposed approach delivers performance that is comparable/better to previous methods (e.g. CoCLR [Han et al., 2020b]) while being much more efficient.
- Thorough ablation studies.

Weaknesses:
- The major weakness of this paper is its limited technical novelty. While the idea of using compressed videos in the context of self-supervised learning is new and interesting, overall the proposed approach is largely built on the ideas from CoCLR [Han et al., 2020b]. Having said this, the proposed approach seems to be both more effective and more efficient than CoCLR. Therefore, I think this would be a good "poster" paper.
- The proposed approach performs quite a bit worse than the recent CVPR 2021 methods such as CVRL, and especially the method of Feichtenhofer et al. "A Large-Scale Study on Unsupervised Spatiotemporal Representation Learning" (CVPR 2021). Having said this, given a very large computational cost of both of these prior methods (i.e. 800 epoch training), I think that it would be unfair to compare the proposed MVCGC approach directly with these methods. The authors should still add the second reference to their related work section, and also to their quantitative comparisons. A discussion on why the direct comparison between these methods is unfair would also be useful.

Questions / Minor Details:
- In L85-88, the authors mention that unlike CoCLR, they construct sample pairs between different views of different clip (i.e., RGB frames of one clip and motion vectors of another clip from the same video). However, the caption of Fig.3 says that the motion vector clip corresponds to the raw RGB clip. Therefore, I was confused, which one is it? Also, if it's the former, could the authors also ablate the latter?
- Typo in Table 3 (it should be Retrieval instead of Retrival).
- In the third part of Table 4, the authors highlight/bold the numbers corresponding to their approach even though CVRL achieves higher accuracy.
- If the authors could include the linear evaluation on Kinetics-400 (similar to "A Large-Scale Study on Unsupervised Spatiotemporal Representation Learning") that would be great. I think the evaluation on UCF and HMDB is a bit saturated and outdated. Therefore, it would be useful to have additional numbers on Kinetics, which is a bit more challenging dataset.

**Time Spent Reviewing:**

2 hours

---

> ### Author Response · Authors · 2021-08-09
> **Response to Reviewer M3h9**
>
> Q1. The major weakness of this paper is its limited technical novelty. While the idea of using compressed videos in the context of self-supervised learning is new and interesting, overall the proposed approach is largely built on the ideas from CoCLR [Han et al., 2020b]. Having said this, the proposed approach seems to be both more effective and more efficient than CoCLR. Therefore, I think this would be a good "poster" paper.\
> A1. Thanks. Our MVCGC is different from CoCLR in two aspects. First, CoCLR decodes compressed videos off-the-fly and computes optical flows (OF) with RGB frames, which is storage- and computation-intensive. In contrast, our MVCGC decodes RGB frames and motion vectors (MV) directly from compressed videos on-the-fly with high efficiency. Second, CoCLR fails to learn the correspondence within the two streams, i.e., there is no information exchange in the pretraining stage, while MVCGC leverages cross guidance to capture mutual information between two streams. Note that our work is the first to exploit contrastive loss in compressed video SSL by leveraging RGB and MV directly from compressed videos. We show that when these two streams are seamlessly integrated by our MVCGC, they carry enough information for video understanding and their fusion is even more powerful than OF+RGB.
>
> Q2. The authors should add the [1] to their related work section, and also to their quantitative comparisons. A discussion on why the direct comparison between these methods is unfair would also be useful.\
> A2. Good suggestion. We will cite Ref. [1] and also add it in quantitative comparisons. Note that it would be unfair to compare our  MVCGC directly with CVRL and Ref. [1]. Our explanations are five-fold: (1) Compressed Videos. CVRL and Ref. [1] utilize raw videos without considering the storage and computation costs during large-scale video self-supervised training. In comparison, MVCGC decodes RGB frames and motion vectors directly from compressed videos and has high efficiency. (2) Backbone. CVRL uses the R3D-50 backbone (33.1M parameters) and Ref. [1] uses the S3D-G backbone (9.1M) which contains an effective gate scheme, while our MVCGC model only uses a weaker backbone S3D (7.9M without gate scheme) following CoCLR. (3) Augmentation. Both CVRL and Ref. [1] use stronger color jittering and temporal augmentation, while our MVCGC follows the standard augmentation in CoCLR. (4) Input resolution. Both CVRL and Ref. [1] use 32 frames with stride 2 of resolution of 224, while MVCGC follows CoCLR by using 32 frames with stride 1 of resolution of 128. (5) Training epochs. CVRL pretrains for 800 epochs and Ref. [1] pretrains for 200 epochs, while MVCGC pretrains for 250 epochs. \
> [1] Feichtenhofer, Christoph, et al. "A Large-Scale Study on Unsupervised Spatiotemporal Representation Learning." CVPR 2021, pp. 3299-3309.
>
> Q3. Confusion In L85-88 and caption of Figure 3.\
> A3. Sorry for the confusion. The sample pairs are constructed between one view of one video clip and the other view of one different video (i.e., RGB frames of one clip and motion vectors of another video, not from the same video).
>
> Q4. Typo in Table 3 and the bold number in Table 4.\
> A4. Thanks for pointing this out. We will make the corrections.
>
> Q5. Include the linear evaluation on Kinetics-400.\
> A5. Good suggestion! The accuracy of linear evaluation on Kinetics-400 is 51.3, better than 50.8 of SeCo [2]. We will add the results in the revised version.\
> [2] Yao, Ting, et al. "Seco: Exploring sequence supervision for unsupervised representation learning." AAAI 2021.

---

> > ### Comment · Reviewer_M3h9 · 2021-08-25
> > **Post-rebuttal Response**
> >
> > Thank you for the response. The rebuttal addresses my concerns. Therefore, I'm keeping my original rating.

---

### Official Review · Reviewer_rcKJ · 2021-07-12

**Rating:** 4
**Confidence:** 4

**Summary:**

This paper presents a self-supervised algorithm for representation learning of videos. The authors propose using RGB clips and Motion Vectors (MV) available in compressed videos to reduce the computational burden of processing. The two streams are seen as views in the contrastive setting: positive pairs are composed of RGB and MV clips of the same video, negative ones have clips belonging to different samples. Moreover, hard positive mining extends the set of positive pairs by seeking other videos that are similar in both views. The approach compares favorably on action datasets UC101 and HMDB51 for both action recognition and video retrieval tasks.

**Limitations And Societal Impact:**

N/A. However, I would argue that no specific discussion is needed on potential negative societal impact for a contrastive learning method.

**Main Review:**

## Strenghts
* The contribution of this paper is timely: the computational requirements of contrastive learning hinders its applicability on video representation learning. Compressed videos enables the shrinkage of required time and storage.
* The work proves how effective use of the less informative MV is on par with the computationally demanding optical flow
* Extensive experiments on action recognition and retrieval prove that the proposed gets competitive results with state of the art methods: linear evaluation and full fine-tuning accuracy for action recognition and R@k metrics for retrieval.

## Weaknesses
* The method builds on [1] by extending the single-view contrastive approach to learn the correspondence between two views and changing the hard-positive condition. However, it is well known that cross-view contrastive is beneficial [2, 3, 4]. Yet, the showed technique for harvesting hard positives is an incremental contribution to the literature.
* How cross guidance and hard positive mining contribute to the final performance is not well disentangled and it's not easy to understand what their impact is. What's the performance loss without hard-positives? What is the effect of using hard positives in the initialization phase? What is the performance of the ensemble of RGB and MV model right after initialization?

### Minor
* Table 2: I find somewhat surprising that the same technique applied to OF gets lower performances, could the authors provide some explanations/insights?
* Line 258: concerning the claim "MVCGC is robust to motion vectors under different encoding configurations": is not really clear how this is proved. In table 2 MV_CoViAr is used for initialization only.
* Line 303: I think that it is "by retrieval with only" RGB "are even higher"
* Appendix, fig 1: the underlying assumptions is that semantically similar videos are close in both RGB and MV space. How do the authors think the model would handle failures of such assumption, e.g., mopping a gymnastic floor, biking in a volleyball arena, etc.?
--------------------
* **Correctness:** The proposed method and the experimental evaluation are sound.
* **Clarity:** The paper is well written and easy to follow. The content is well organized: contribution is expressed concisely and the experimental part is well explained.
* **Relation to prior work:** Authors relates their work with previous literature while concisely highlighting their contribution.
* **Reproducibility:** Architecture, method and optimization hyperparameters are available for implementation, missing augmentation parameters (random cropping, blurring, jittering) and missing code limits the reproducibility of the work though.
* **Originality**: The contribution of this paper is incremental with respect to cited papers, the real advantages of the technique are not thoroughly highlighted in the ablation part.
------------------------
[1] Han, Tengda, Weidi Xie, and Andrew Zisserman. "Self-supervised Co-Training for Video Representation Learning." Advances in Neural Information Processing Systems 33 (2020): 5679-5690.

[2] Tian, Yonglong, Dilip Krishnan, and Phillip Isola. "Contrastive multiview coding." Computer Vision–ECCV 2020: 16th European Conference, Glasgow, UK, August 23–28, 2020, Proceedings, Part XI 16. Springer International Publishing, 2020.

[3] Bachman, Philip, R. Devon Hjelm, and William Buchwalter. "Learning Representations by Maximizing Mutual Information Across Views." Advances in Neural Information Processing Systems 32 (2019): 15535-15545.

[4] Chen, Ting, et al. "A simple framework for contrastive learning of visual representations." International conference on machine learning. PMLR, 2020.


**Time Spent Reviewing:**

3

---

> ### Author Response · Authors · 2021-08-09
> **Response to Reviewer rcKJ**
>
> Q1. The method builds on [1] by extending the single-view contrastive approach to learn the correspondence between two views and changing the hard-positive condition. However, it is well known that cross-view contrastive is beneficial [2, 3, 4]. Yet, the showed technique for harvesting hard positives is an incremental contribution to the literature.\
> A1. We do not agree. Our main contribution lies in that, for the first time, we exploit how to solve self-supervised video representation learning in both effective and efficient ways. We achieve this goal by leveraging the MV stream in compressed videos as an extra stream in our proposed contrastive learning algorithm. Although contrastive learning has also been shown to be effective in recent approaches such as CMC [2] and CoCLR [1], our MVCGC is different from these approaches in how contrastive loss and sample pairs are defined. First, differing from CMC which only mines a single positive (regarding the hard positives as negative samples), MVCGC incorporates the hard-positive mining procedure. Second, the co-training in CoCLR fails to learn the correspondence between two views (i.e., there is no information exchange in the pre-learning stage), but our MVCGC leverages the cross-guidance to capture mutual information between two streams. We will clarify this in the revised version.
>
> Q2. How cross guidance (CG) and hard positive mining (HPM) contribute to the final performance is not well disentangled and it's not easy to understand what their impact is. What's the performance loss without hard positives? What is the effect of using hard positives in the initialization phase? What is the performance of the ensemble of the RGB and MV models right after initialization?\
> A2. To answer these questions, we have conducted additional experiments and obtained more results in the following table. Note that we can't obtain the results of using hard positives in the initialization stage: given a target sample, finding similar samples (i.e., hard positives) based on a randomly initialized encoder is meaningless.
>
> | Method   | Pretrain | Linear Probe  | Retrieval |
> | ----------- |:-----------:|:-----------:|:-----------:|
> | Init. (RGB) | RGB |52.3|33.1|
> | Init. (Ensemble) | RGB+MV|67.0|34.2|
> | Init. + CG |RGB+MV|71.3|59.9|
> | Init. + CG + HPM|RGB+MV|**73.1**|**60.8**|
>
> It can be seen that: (1) The ensemble of the RGB and MV models outperforms the single RGB-only model. (2) CG leads to a large improvement over the ensemble model, showing the effectiveness of cross guidance. (3) The comparison Init. + CG + HPM vs. Init. + CG further shows the contribution of hard positive mining.  We will add the mentioned results in the revised version.
>
> Q3. In Table 2, why the same technique applied to OF gets lower performances?\
> A3. Table 2 shows the thorough comparison between utilizing MV and OF. Although OF is better than MV when they are used alone, MV can be combined with RGB seamlessly, and thus MV+RGB performs even better than OF+RGB (see Table 2). This is a little counter-intuitive but can be explained. Note that MV and RGB are two original components of video codecs and MV+RGB carries more raw video information than OF+RGB. This enables our MVCGC to better learn the correspondence within the two streams (i.e., MV and RGB), leading to higher accuracy and efficiency.
>
> Q4. Line 258: the claim "MVCGC is robust to motion vectors under different encoding configurations": is not clear how this is proved.\
> A4. Sorry for the confusion. We show the experiment results of pretraining on Kinetics-400 and then transferring to UCF/HMDB in Table 4. Although the compression settings in these datasets are different, our MVCGC still achieves state-of-the-art results, indicating its robustness under different compression settings. We will clarify this.
>
> Q5. I think the sentence in Line 303 should be "...by retrieval with only" RGB "are even higher...".\
> A5. Thanks for pointing this out. This full sentence should be "Particularly, our obtained retrieval results by retrieving with both RGB and MV are even higher than those of the latest competitor CoCLR which uses both RGB and OF for retrieval, indicating the effectiveness of our MVCGC".
>
> Q6. Appendix, fig 1: the underlying assumption is that semantically similar videos are close in both RGB and MV space. How do the authors think the model would handle failures of such assumption, e.g., mopping a gymnastic floor, biking in a volleyball arena, etc.?\
> A6. Good suggestion! First, we use the intersection operation instead of the union one according to the ablation results in Table 3. Second, we have two solutions to handle the mentioned failures. (1) The hyper-parameter k is adjusted to enable different streams to have different k, i.e., k_{MV} is set to be larger than k_{RGB}. (2) The hard-positive samples are defined as the union of the intersection set and the set of top-k similar samples in the MV stream. All similar actions are thus taken as positive samples.

---

> ### Author Response · Authors · 2021-08-23
> **Looking forward to your post-rebuttal feedback**
>
> Dear Reviewer rcKJ,
>
> Thanks again for your insightful suggestions and comments. As the deadline for discussion is approaching, we are happy to provide any additional clarifications that you may need.
>
> In our previous response, we have carefully studied your comments and made detailed responses summarized below:
> 1. Clarified that the proposed algorithm is different from other cross-view contrastive methods in two aspects.
> 2. Conducted additional experiments to study the impact of CG and HPM and verify the effectiveness of our framework.
> 3. Explained why the MV outperforms OF when trained together with RGB.
> 4. Proven why MVCGC is robust to MV under different encoding configurations.
> 5. Clarified some minor confusion.
> 6. Discussed the modifications of our designed model when handling failures.
>
> We hope that the provided new experiments and additional explanations have convinced you of the merits of our submission.
>
> Please do not hesitate to contact us if there are other clarifications or experiments we can offer. Thank you very much!
>
> Thank you for your time!
>
> Best,\
> Authors

---

### Official Review · Reviewer_ZVsq · 2021-07-15

**Rating:** 6
**Confidence:** 5

**Summary:**

This paper proposes a self-supervised method for compressed video representation learning, which helps save both storage and computation.

**Limitations And Societal Impact:**

The reviewer would encourage the authors to discuss limitations and impacts of the proposed method, as mentioned in the 'Main Review'.

**Main Review:**

Strength:

- The proposed method directly uses RGB and motion vectors in the compressed video for efficient representation learning.
- Experimental results show that the proposed method reduces the total (inference) time and storage, and achieves marginal improvement compared to state-of-the-art methods.

Weakness:

- Adding temporal learning objectives such as changing the temporal order of RGB frames or motion vectors within a clip ( as in IMRNet ), which is missing from the proposed method, would greatly help learning video representation.
- Modeling of the proposed method mostly applies the InfoNCE loss, only selecting of positive and negative samples are updated for RGB and motion vectors.
- As shown in table 1, the computation overhead largely comes from the preprocessing time. Though not an apple-to-apple comparison, parallel re-encoding for IMRNet may largely mitigate the overhead?

**Time Spent Reviewing:**

4.5 hrs

---

> ### Author Response · Authors · 2021-08-09
> **Response to Reviewer ZVsq**
>
> Q1. Adding temporal learning objectives is missing from the proposed method, which would greatly help to learn video representation.\
> A1. Good suggestion! In this work, we focus on exploits contrastive learning in compressed videos by using motion vectors as an extra input stream. Recent works (e.g., IMRNet or Pace) design pretext tasks of changing or accelerating the temporal order, following another way of self-supervised learning. According to Table 4, our MVCGC outperforms these competitors by exploiting only contrastive learning, showing the effectiveness of our proposed method. To further show the high flexibility of our MVCGC, we will include the results of adding temporal learning objectives in the revised version.
>
> Q2. Modeling of the proposed method mostly applies the InfoNCE loss, only selecting positive and negative samples are updated for RGB and motion vectors.\
> A2. Sorry for the confusion. Our main contribution lies in that, for the first time, we exploit how to solve self-supervised video representation learning in both effective and efficient ways. We achieve this goal by leveraging the MV stream in compressed videos as an extra stream in our proposed contrastive learning algorithm. Although contrastive learning has also been shown to be effective in recent approaches such as CMC and CoCLR, our MVCGC is different from these approaches in how contrastive loss and sample pairs are defined. First, differing from CMC which only mines a single positive (regarding the hard positives as negative samples), MVCGC incorporates the hard-positive mining procedure. Second, the co-training in CoCLR fails to learn the correspondence between two views (i.e., there is no information exchange in the pre-learning stage), but our MVCGC leverages the cross-guidance to capture mutual information between two streams. We will clarify our main contribution more precisely.
>
> Q3. As shown in Table 1, the computation overhead largely comes from the preprocessing time. Though not an apple-to-apple comparison, parallel re-encoding for IMRNet may largely mitigate the overhead?\
> A3. Yes, parallel re-encoding may be a choice. However, it still has three drawbacks: (1) As long as the re-encoding process exists, the pre-processing time of IMRNet is longer than that of MVCGC. (2) Considering that the computation overhead difference between IMRNet and MVCGC comes from not only the pre-processing stage (0.128-0.008=0.12ms) but also the inference stage (0.227-0.0703=0.1567ms). Parallel re-encoding doesn't address the main efficiency issue. (3) The large storage burden problem of IMRNet (6.1GB for IMRNet vs 1.9GB  for MVCGC) could not be solved by parallel re-encoding.

---

> > ### Comment · Reviewer_ZVsq · 2021-09-01
> > **Post-rebuttal Comments**
> >
> > Many thanks to the insightful reviews from all reviewers.
> >
> > After carefully reading the authors' rebuttal, I think it addressed my concerns, and I'm keeping my original rating for this paper.

---

### Official Review · Reviewer_Rmsi · 2021-07-16

**Rating:** 7
**Confidence:** 4

**Summary:**

The authors propose a self-supervised learning (SSL) method for learning strong visual representations based on compressed videos. Compared with prior work, which sees videos in RGB only, the proposed method is more efficient and leverages the motion information that's already accessible in compressed videos. The authors propose to use a contrastive learning method to supervise on kind of modality with correspondence of another modality. On multiple datasets (mainly UCF and HMDB), good performance is achieved, with both linear probing and transfer learning settings.


**Limitations And Societal Impact:**

1. In the paper, the authors perform experiments/evaluation solely on videos. I was wondering if the method learns good representations for "image" tasks as well. Namely, leveraging MV can potentially give us better RGB representations for images, too. I wonder what the authors think about this.
2. If the hard-positive mining is based on the learned features, would there be any "chichen-and-egg" problem introduced by the proposed procedure?
3. I like the comparison to MV_{CoViAR}, but I wonder how each specific compression setting affects the results. For example, how does GOP size, the use of B-frame vs P-frame, etc. affect results.

Regarding societal impact, I don't see anything particularly concerning (or more concerning) than other SSL work.

**Main Review:**

[originality: medium-high]
 - The idea of using compressed videos for self-supervised learning is novel/original. Most prior work that operates on compressed videos focues on supervised learning setting only. The implementation of this idea is standard/expected (this is not a drawback).

[quality: medium-high]
 - The paper covers useful ablation study that I'd expect. In particular, the authors demonstrate the usefulness of the compressed-video modality (MV) compared to RGB only (Table 3). Compared to prior work, the performance is also strong while being significantly more efficient. The ablation study that compares with optical flow and low-res optical flow is also interesting (Table 2).

[clarity: high]
 - I think this paper is overall clearly writen. One question about implementation details: How is the similarity calculated in the hard-positive mining procedure?

[significance: medium-high]
 - The area of SSL draws a lot of attention in our community and is of great potential. I think this work extends our knowledge from a new dimension with the use of compressed videos. I think this work can lead to promising new directions in SSL research.




**Time Spent Reviewing:**

1.5 hours

---

> ### Author Response · Authors · 2021-08-09
> **Response to Reviewer Rmsi**
>
> Q1. How is the similarity calculated in the hard-positive mining procedure?\
> A1. Sorry for the confusion. As shown in Figure 3, we follow MoCo, where two queues (one for RGB and one for MV) containing negative keys from recent batches are maintained. MoCo follows the assumption that the target sample is semantically different from all the other samples in the queue. This is counter-intuitive, since different samples may also have similar semantic content. In this work, we thus add the hard-positive mining procedure into MoCo. Specifically, given a target sample, MVCGC first calculates the cosine similarity between its RGB embedding and all other RGB embeddings in the queue. The top-k similar samples in the queue are regarded as semantically similar to the target sample in the RGB stream. Moreover, MVCGC also calculates the cosine similarity of MV embeddings between the target sample and all other samples in the queue to obtain top-k similar semantically similar samples in the MV stream. The final hard-positive samples are mined by taking the intersection between the obtained results of two streams. The hyper-parameter k is set to 5 according to the ablation study in Table 3. We will clarify this in the final version.
>
> Q2. I was wondering if the method learns good representations for "image" tasks as well. Namely, leveraging MV can potentially give us better RGB representations for images, too.\
> A2. Yes, Table 3 indeed shows that leveraging MV can give us significantly better RGB representations for images. For easier understanding, the corresponding results are drawn from Table 3  as follows:
>
> | Pre-training   | Evaluation | Linear Probe  | UCF Retrival |
> | ----------- |:-----------:|:-----------:|:-----------:|
> | RGB		|RGB |52.3|33.1|
> | RGB+MV	|RGB |**73.1**|**60.8**|
>
> Q3. If the hard-positive mining is based on the learned features, would there be any "chicken-and-egg" problem introduced by the proposed procedure?\
> A3. Note that solving the "chicken-and-egg" problem has been considered in our procedure. Specifically, our MVCGC proceeds in two stages: the initialization stage and the cross-guidance stage (line 206). In the first stage, the RGB encoder and MV encoder are initialized by minimizing the classical contrastive loss. With this initialization, we can then perform the hard-positive mining procedure. Concretely, in the second stage, the two encoders are cross-trained to minimize our proposed contrastive loss in Eq. (5). Under the cross-guidance of two streams, the mined hard-positive samples are supposed to be reliable. Results in Table 3 indicate that: (1) Both the initialization stage and cross-guidance stage benefit the video representation learning; (2) The initialization stage does help to overcome the "chicken-and-egg" problem. For easier understanding, the corresponding results are drawn from Table 3  as follows:
>
> | Method   | Linear Probe  | UCF Retrival |
> | ----------- |:-----------:|:-----------:|
> | Init. |52.3|33.1|
> | cross-guidance |65.0|48.4|
> | Init. + cross-guidance |**73.1**|**60.8**|
>
> Q4. How each specific compression setting (GOP size, the use of B-frame vs P-frame) affects the results in the comparison to MV_{CoViAR}.\
> A4. In the comparison to MV_{CoViAR}, our MVCGC follows the same compression setting as in the CoViAR's paper: the GOP size = 12, no B-frame. In our other experiments, MVCGC adopts the specific compression setting of the original videos in the corresponding datasets. We show the experiment results of pretraining on Kinetics-400 and then transferring to UCF/HMDB in Table 4. Although the compression settings in these datasets are different, our MVCGC still achieves state-of-the-art results, indicating its robustness under different compression settings.

---

> > ### Comment · Reviewer_Rmsi · 2021-08-28
> > **Re: Response to Reviewer Rmsi**
> >
> > Dear authors,
> >
> > Thanks for providing the author feedback!
> >
> > My questions regarding "similarity calculation" is addressed. I'd suggest adding some of these descriptions to paper.
> >
> > Regarding Q2, if I understand correctly, the results you point to here are still on video datasets. I was mainly asking how the RGB representation performs on "image" datasets, e.g. ImageNet. I understand that this paper mainly focuses on videos, so not including those results is not a weakness, but thought including that would be interesting.
> >
> > Regarding Q3, makes sense! Thank you.
> >
> > Regarding Q4, I still think knowing how each specific compression setting affects the results would be interesting. It would also help us understand the robustness of the proposed method.
> >
> > Overall, my main questions are answered and I keep my recommendation unchanged.

---

### Author Response · Authors · 2021-09-02
**Summary of Our Rebuttals and Discussions**

Dear ACs and Reviewers,

We sincerely appreciate your time and efforts in reviewing our paper. We have made detailed responses to the constructive comments of all reviewers. The corresponding discussions will be delivered in our final version.

During the rebuttal period, we received positive responses (7-6-6) from Reviewers Rmsi, ZVsq & M3h9. However, even with two rebuttals posted, we still didn't receive any responses from Reviewer rcKJ. With new experiments and additional explanations, we believe that all concerns of Reviewer rcKJ have been addressed. We hope that the ACs could make the final decision based on these facts.

Best,\
Authors

---

### Decision · Program_Chairs · 2021-09-28

**Decision:**

Accept (Poster)

**Comment:**

The paper received the following final ratings: 6, 6, 7, 4.

Reviewer rcKJ raised concerns about the close technical reliance on CoCLR [Tengda et al., NeurIPS 2020] and the lack of understanding about the impact of cross guidance on performance. The authors rebutted these points in their response. Unfortunately, Reviewer rcKJ failed to provide post-rebuttal comments, but the ACs found the points made by the authors convincing.

The other reviewers appreciated the responses provided by the authors, especially the clarification about the similarity calculation, the differences wrt CoCLR, and the inclusion of the linear evaluation on Kinetics-400. As noted by Reviewer Rmsi, an ablation on how the different compression settings affect performance would further strengthen this work. The ACs agree with the majority recommendation of acceptance expressed by the reviewers.

**Consistency Experiment:**

NeurIPS has a long history of experimentation. In 2014, NeurIPS ran an experiment in which 10% of submissions were reviewed by two independent committees to quantify the randomness in the review process. This year, we repeated a variant of this experiment to see how the quality of the review process has changed over time.  This paper was part of the experiment and was therefore assigned to two committees (consisting of reviewers, an Area Chair, and a Senior Area Chair) that reached independent decisions.  If both committees made the same recommendation, this recommendation was followed. If a single committee recommended acceptance, the paper was accepted (with the exception of a few cases in which the other committee identified what we considered a fatal flaw, e.g., an error in a key result).

This copy’s committee reached the following decision: **Accept (Poster)**

The other committee assigned to the paper recommended **Reject**.  You can find the other set of reviews, along with any follow up discussion with the authors here:
https://openreview.net/forum?id=RdWt-VDPZEG